# Validity of the Friedrich Short Form of the Questionnaire on Resources and Stress in Parents of Individuals with Autism Spectrum Disorder

**DOI:** 10.3390/ijerph182212174

**Published:** 2021-11-19

**Authors:** Eun-Young Park

**Affiliations:** Department of Secondary Special Education, Jeonju University, Jeonju 55069, Korea; eunyoung@jj.ac.kr; Tel.: +82-63-220-3186

**Keywords:** Friedrich short form of the Questionnaire on Resources and Stress, parents, autism spectrum disorder, validity, caregiving burden

## Abstract

There is insufficient knowledge about the psychometric properties of the Friedrich short form of the Questionnaire on Resources and Stress (QRS-F) used to measure the caregiving burden of caregivers of individuals with autism spectrum disorder (ASD). The present study, therefore, aimed to confirm the validity of the QRS-F. The data selected using the systematic sampling method were analyzed to verify the factor structure of the QRS-F on parents of individuals with ASD. Exploratory and confirmatory factor analyses were employed to confirm the validity and the factor structure of the scale. The Pearson correlation coefficient was calculated to verify the relation with other measures. The original factor model was not appropriate to assess the caregiving burden on parents of individuals with ASD because the models did not show adequate fit indices. The evaluation of results based on a total score was explored, which demonstrated the expected association between depression severity and caregiving time. Overall, this study supports the use of the QRS-F for measuring the caregiving burden of parents of individuals with ASD by comparing the total score with other related variables.

## 1. Introduction

Caring for a child is a typical parental role, yet the excessive demands associated with caring for a child with disabilities can lead to increased burden or stress [1,2]. Looking after children with chronic conditions has often been reported to be associated with negative health outcomes, such as depression, stress, anxiety, and low self-efficacy for the caregivers [1,3,4,5,6,7]. The caregiving burden causes psychological changes, including depression, insomnia, and loss of motivation [8,9]. Clyburn et al. [10] also found that the burden on caregivers results in depression and related symptoms. Furthermore, the caregiving burden is known to reduce subjective psychological well-being and life satisfaction in caregivers [11,12]. Moreover, caregiving could affect the overall health of caregivers. A Canadian-population-based study showed that caregivers of children with health problems were more than twice as likely to develop chronic conditions, activity limitations, as well as elevated depressive symptoms, and showed greater odds of suffering from poor general health than caregivers of healthy children [13].

Autism spectrum disorder (ASD), the most serious and complex condition among neurodevelopmental disorders, is a universal overall developmental disorder characterized by limited and repetitive behaviors or interests, along with defects in social communication [14]. In 2000, the revised Diagnostic and Statistical Manual of Mental Disorders (DSM-IV-TR) defined ASD as a qualitative impairment in social interaction and communication, with restricted, repetitive, and stereotyped behaviors, interests, and activities, and an onset prior to 3 years of age [15]. The recently modified version, the DSM-5, includes these criteria; however, the triad of diagnostic criteria from the DSM-IV-TR was replaced in the DSM-5 with two composite diagnostic criteria: (1) impairments in social communication and (2) restricted, repetitive behavior [16].

Individuals with ASD are likely to exhibit clinical-level emotional behavior problems, as well as major symptoms, such as difficulty in social communication and forming relationships [17]. Therefore, they require assistance with daily activities, and it is especially difficult for them to lead independent lives. Parents of children with ASD are responsible for managing and nurturing them, and experience significantly higher parenting stress and emotional distress than parents of children with other disabilities [18,19]. Mothers of children with ASD experience higher parenting stress than those of children with other developmental disorders, such as Down syndrome and intellectual disabilities [20,21,22]. In addition to parenting stress, mothers of children with ASD experience significantly higher levels of emotional stress, depression, anxiety, and anger than those of children with other disabilities, and are highly prone to reduced physical health due to lack of sleep and chronic fatigue [23]. These findings suggest that caregivers of individuals with ASD find it more challenging to maintain an emotionally and physically healthy life than those of persons with other mental disorders. They also suffer considerable burden and exhaustion in the process of raising children with ASD who have overall functional impairment.

Since caring for a family member with a disability has a well-known effect on the health of the caregiver, it is important to regularly determine factors that induce burden/stress on the caregiver [6]. This can only be achieved by measuring psychometrically sound outcomes [24]. To provide support and services to caregivers, it is necessary to manage the care burden imposed on the caregiver, and this can only be accomplished if there is an appropriate measure to assess the caregiver’s burden [2,7,25].

The Zarit Burden Scale (ZBS) is a frequently used measure of caregiver burden for caregivers of children with ASD. Consisting of 29 questions, it was developed to measure the burden of the family or spouse caring for elderly dementia patients [26]. Furthermore, the ZBS was employed by Pandey et al. [27] to demonstrate that levels of education, anxiety, and depression were significantly associated with the burden of caregiving among caregivers of children with ASD. Another scale for measuring caregiving burden in families of individuals with ASD is the 21-item Caregiver Strain Questionnaire, which was developed by Brannan et al. [28] to measure the strain of caring for children with emotional or behavioral disorders. The Friedrich short form of the Questionnaire on Resources and Stress (QRS-F) [29] is frequently used to study stress in families of children with disabilities [30] and is widely employed for measuring the caregiving burden in families of children with ASD [31,32,33,34]. However, there is limited knowledge regarding its psychometric properties when measuring the burden among caregivers of individuals with ASD. To ascertain this, it is necessary to study its validity and reliability in detail. Therefore, the purpose of this study was to evaluate the psychometric soundness of the QRS-F when applied to caregivers of individuals with ASD.

## 2. Materials and Methods

### 2.1. Study Data

Data from the 2011 survey on the Actual Conditions of Individuals with Developmental Disabilities, conducted by the Ministry of Health and Welfare, were used for this study. The population of this survey comprised a list of individuals with developmental disabilities registered on the Ministry’s database as of August 2011. The number of those with ASD totaled 15,498. The stratifying sampling method was used to collect data from 1500 persons with intellectual disabilities and ASD for the survey. Among them, data from 293 persons with ASD were used for analysis. The specific characteristics of individuals with ASD and that of their parents are shown in Table 1. The stratification variable is based on the type and grade of disability, of which characteristics in the population with developmental disabilities are discriminative, and the representativeness of people with developmental disabilities in Korea was secured by adding districts. In Korea, the diagnosis and grade of the disability by a doctor are needed for registration. Regarding the grade of disability, Grade 1 individuals with ASD, or those who were assessed to have a pervasive developmental disorder according to the diagnostic criteria of ICD-10 (International Statistical Classification of Diseases and Related Health Problems-10), who do not show normal development, possess an IQ of 70 or less and a Global Assessment Scale for Developmentally Disabled (GAS) score of 20 or less due to impairment of function and ability were the most common (39.6%). Grade 2 individuals with ASD, or those who were assessed to have a pervasive developmental disorder according to the diagnostic criteria of ICD-10, who do not show normal development, possess an IQ of 70 or less and a GAS score of 21–40 due to impairment of function and ability comprised 33.1%. Grade 3 individuals with ASD, or those who were assessed to have a pervasive developmental disorder according to the diagnostic criteria of ICD-10, who do not show normal development, and had an IQ of 71 or above, and GAS score of 41–50 due to impairment of function and ability comprised 27.3%.

The number of observations required to show the reliability of factors depends on the data. Furthermore, the variables bare load on the different factors. A factor is reliable if it has 10 or more variables with loadings of 0.4 and *n* ≥ 150, and if it has factors with only a few loadings, it requires *n* ≥ 300 [35]. A total sample size of approximately 300 was considered as good and that of around 200 as fair relative to the total sample size [36].

### 2.2. Measurements

We used the QRS-F to measure the burden of care among parents of children with ASD. The QRS-F was developed by Holroyd [37] to measure the burden of caring for children with chronic diseases [31]. This 52-item scale measures the caregiving burden resulting from problems faced by parents. This scale consists of four factors such as the wider family, the parents’ pessimistic attitude concerning their children, the child’s characteristics, and the child’s physical incapacities, with item-total correlations ranging from 0.15 to 0.63, and a mean inter-item correlation of 0.26. The response could be “yes” or “no” for each item. The total score was calculated by adding 1 for every “yes” response; a high score indicates a high level of caregiving burden. In this study, the reliability of this tool was 0.826, thus making it suitable to use for the desired measurement.

The Center for Epidemiologic Studies Depression (CES-D) is an 11-item scale that was used in this study. Each item was measured using the Likert scale (0: mostly not, 1: moderately, 2: mostly yes, 3: always). Questions 2 and 7 were used to measure positive emotions and were emphasized during the scoring process. The scores of all items are summed, and a higher score indicates more depressive symptoms. The results of the confirmatory factor analysis of this scale used in a survey on older adults receiving home healthcare in the study by Gellis [38] were similar to those obtained using the original scale developed by Radloff [39]. In a study involving adolescents, the internal agreement (Cronbach’s α) of the CES-D scale was reported as 0.909 [40].

The severity of the following 14 behavior problems: delusion, illusion, aggressive behavior, depression, anxiety, mania, unconcern, sexual or impulsive behavior, screams/swearing or obscenity, restricted repetitive behavior, sleep problems, problems regarding eating habits, kleptomania, and wandering, was assessed. The six-point Likert scale, with responses ranging from 0 (never) to 5 (very severe), was used to obtain the total rating, which was entered into a path model. In a previous study, it was reported that this measure had a reliability of 0.887 [41].

In this study, the caregiving time per day was measured, and the daily average of the time consumed to care for individuals with ASD was used for the analysis.

### 2.3. Statistical Analysis

The main method for examining the construct of a measurement tool is factor analysis, which is a statistical approach used to test the adequacy of a conceptual model. Exploratory factor analysis (EFA) is used to determine the underlying structure of related variables. Confirmatory factor analysis (CFA) may be used to investigate whether the established dimensionality and factor-loading pattern fits a sample from a new population. CFA is a type of factor analysis that is used to test the hypothesis that there is a fundamental relation between observed variables and potential latent constructs. To confirm the validity and factor structure, both EFA and CFA were employed. First, CFA was performed to calculate the model fit indices of a four-factor model, which was proposed in a previous study [31]. Then, EFA was performed to identify the number of factors and factor loadings. To detect the suitability of the data for the EFA, the Kaiser–Meyer–Olkin measure was calculated. The value was 0.866, thus indicating its suitability. Bartlett’s sphericity test was used to reject the null hypothesis that there was no factor structure with a significance level of *p* < 0.000. The main axis factor analysis was performed using IBM SPSS Statistics for Windows, version 23 (IBM Corp., Armonk, NY, USA), and the oblique rotation was performed using the Promax method. The appropriateness of the model was verified by several fit indices in the CFA [42]. Fit indices were classified as incremental (comparative fit index (CFI), Tucker–Lewis Index (TLI), and normed fit index (NFI)), and absolute (chi-square and root mean square error of approximation (RMSEA)). RMSEA values < 0.05, 0.06–0.08, 0.08–0.10, and > 0.1 indicate a good, reasonable, mediocre, and poor fit, respectively. NFI, CFI, and TLI values > 0.90 also indicated a good fit. The Analysis of Moment Structure, version 20.0 statistical program, was used to perform the CFA for obtaining maximum-likelihood estimates of the model parameters and goodness-of-fit indices. Convergent validity was verified by using correlation coefficients with the CES-D score, problem behavior severity, and caregiving time. SPSS 25.0 (IBM Corp. New York, NY, USA) and AMOS 25.0 (IBM Corp. New York, NY, USA) were used for statistical analyses.

## 3. Results

### 3.1. Confirmatory Factor Analysis

The four-factor model proposed in a previous study [37] was verified for parents of individuals with ASD. This model is not appropriate to assess the caregiving burden in parents of individuals with ASD because the goodness-of-fit index values obtained were NFI = 0.543, TLI = 0.670, CFI = 0.685, and RMSEA = 0.065.

### 3.2. Exploratory Factor Analysis

To identify the factors affecting the burden of support, principal component analysis was performed with an eigenvalue of one. The eigenvalues, explanatory variances, and cumulative explanatory variances of each extracted factor are shown in Table 2. The results showed that there were 13 factors with no distinct factor structure.

Table 3 presents the results of the rotated factor matrix based on 13 factors derived from the EFA analysis and the results of Holroyd [37], who proposed a four-factor structure of the QRS-F. The results of the analysis based on the factor structure presented at the time of the QRS-F development and the survey data of the parents of individuals with ASD were found to be inconsistent.

### 3.3. Convergent Validity

Three sets of analyses were conducted to explore the convergent validity of the total score obtained using the QRS-F. Correlations were explored among the QRS-F scores, CES-D scores, problem behavior severity, and caregiving time (Table 4). The results showed significant correlations among the scores. The range of the correlation coefficient ranged from 0.212 to 0.737. The QRS-F total score showed the lowest coefficient with caregiving time (0.212) and the highest coefficient with problem behavior severity (0.300).

## 4. Discussion

This study aimed to confirm the psychometric properties of the QRS-F to assess the caregiving burden of parents of individuals with ASD and to analyze the validity of the QRS-F factor structure. The findings suggest that the construct validity of the four-factor model is not adequate, and only the QRS-F total score is a useful measure in assessing the caregiving burden of parents of individuals with ASD.

The original validation study, which was performed to verify the psychometric properties of the QRS-F in parents of young children with ASD, reported the Kuder–Richardson–20 reliability coefficient for the questionnaire as 0.95. The Kuder–Richardson coefficients for the mothers of children with ASD were reported as 0.85 and 0.93, and 0.88 was reported for the fathers of children with ASD in the total score based on 31 QRS-F items [33]. The results of our study showed good indices of internal consistency (0.826). Furthermore, the earlier study also suggested that the four-factor model is not suitable for measuring the caregiving burden in parents of individuals with ASD, which is in line with this study’s finding [33].

Individuals with ASD are more likely to behave outside the rules and norms in public places and express sudden behaviors and emotions [43] that caregivers are unable to control. Thus, caregivers experience helplessness, anxiety, and embarrassment, which leads them to avoid going out or participating in social activities [44]. Moreover, it is difficult for parents to be exposed to new stimuli or environments because of their child’s preference for and preoccupation with the same activities, interests, and certain rules that children with ASD favor; therefore, the parents may be restricted from exposure to new experiences [45]. Previous studies showed that mothers of children with ASD are socially isolated and experienced low self-esteem and depression [46,47]; their quality of life was also significantly lower [48]. The parenting burden of mothers of children with developmental disabilities is a strong predictor of depression [41,49]. In particular, parents who are the primary caregivers experience caregiving burdens, such as feelings of failure or frustration repeatedly throughout their lives [50]. Parents raising children with developmental disabilities easily feel physically and mentally fatigued because there are many instances when they experience anger or aggravation as a result of their situation, with the resulting stress having a negative effect on their mental health [51].

While the support of the social system is important to enable individuals with ASD to lead independent and comfortable lives, the continuous care provided by parents and caregivers is all the more crucial. To minimize the negative impact on the whole family due to the burden of caring for individuals with ASD, it is necessary to form a sustainable and practical support system that can alleviate the caregiving burden. Furthermore, it is necessary to accurately measure the level of caregiving burden to prepare a support program and verify its effectiveness. We showed that the total score obtained using the QRS-F is beneficial for measuring the caregiving burden experienced by parents of individuals with ASD.

There was one main limitation to this study: a small sample size. Although the sample size was considered fair to good for the EFA, it was still not sufficient. Further studies using bigger sample sizes are needed to more accurately verify the QRS-F factor structure.

## 5. Conclusions

In this study, the validity of the QRS-F was evaluated using factor analysis among the parents of individuals with ASD. Our results showed that the four-factor model proposed in the original study is not appropriate for measuring the caregiving burden in parents of individuals with ASD. The unidimensionality of the QRS-F was confirmed through the results of the CFA. According to the result of this study, the total score obtained using the QRS-F is recommended and the comparison of the sub-factor score is not recommended. The convergent validity was confirmed through this study. Therefore, the QRS-F can be used to evaluate the total level of the caregiving burden on parents of individuals with ASD, and the total score can be compared with other related variables.

## Figures and Tables

**Table 1 ijerph-18-12174-t001:** Characteristics of individuals with ASD and their caregivers.

Characteristics	Frequency	%
Individuals with ASD		
Sex		
Male	245	83.6
Female	48	16.4
Age (years)		
~6	17	5.8
7~18	184	62.8
19~	92	31.4
Grade of disability		
1	116	39.6
2	97	33.1
3	80	27.3
Caregivers		
Sex		
Male	47	16.0
Female	246	84.0
Age (years)		
≤39	66	22.5
40~49	142	48.5
50~59	65	22.2
60+	20	6.8

Note: ASD = autism spectrum disorder.

**Table 2 ijerph-18-12174-t002:** The exploratory factor analysis result of the QRS-F.

Factors	Initial Eigenvalue		
	Total	% Variance	Cumulative %
1	10.333	19.871	19.871
2	5.094	9.795	29.666
3	2.330	4.480	34.146
4	2.173	4.179	38.325
5	1.715	3.298	41.623
6	1.516	2.915	44.538
7	1.489	2.863	47.400
8	1.278	2.459	49.859
9	1.260	2.423	52.282
10	1.155	2.221	54.503
11	1.125	2.163	56.666
12	1.045	2.010	58.676
13	1.016	1.953	60.629

**Table 3 ijerph-18-12174-t003:** The rotated factor matrix based on 13 factors and 4 factors by Holroyd.

Category	Factors Based on the EFA Analysis													Factors by Holroyd [37]			
	1	2	3	4	5	6	7	8	9	10	11	12	13	F1	F2	F3	F4
No.26	0.83	0.37	−0.07	−0.06	0.39	0.09	0.20	0.05	0.35	−0.01	−0.04	0.07	0.00				0.70
No.28	0.82	0.41	−0.06	−0.08	0.33	0.12	0.23	0.10	0.29	0.00	−0.07	0.03	−0.12		0.59		
No.24	0.82	0.37	−0.05	−0.08	0.32	0.05	0.23	0.14	0.38	0.03	−0.02	0.01	−0.02	0.62			
No.25	0.78	0.48	−0.07	−0.17	0.32	0.24	0.20	0.10	0.30	−0.07	−0.10	−0.02	0.00		0.40		
No.30	0.68	0.63	−0.12	−0.06	0.40	0.16	0.15	0.19	0.17	−0.09	−0.03	0.02	−0.15			0.56	
No.22	0.61	0.44	0.05	0.08	0.32	−0.06	0.27	0.04	0.19	0.10	0.03	0.25	−0.07		0.45		
No.21	0.60	0.28	−0.03	0.15	0.29	0.05	0.50	0.18	0.45	0.08	0.01	0.01	0.04			0.43	
No.31	0.54	0.53	−0.04	−0.19	0.42	0.32	0.23	0.03	0.38	0.14	−0.22	0.08	−0.04	0.48			
No.19	0.50	0.19	−0.01	0.11	0.16	0.11	0.49	0.34	0.21	0.07	0.04	0.03	−0.20			0.60	
No.6	0.49	0.74	−0.03	−0.23	0.38	0.24	0.19	0.02	0.25	−0.13	−0.10	0.04	−0.06			0.40	
No.9	0.46	0.73	−0.13	−0.11	0.45	0.27	0.29	−0.03	0.31	−0.23	−0.03	0.21	0.03	0.63			
No.11	0.42	0.67	−0.18	−0.27	0.51	0.32	0.09	0.36	0.43	−0.04	−0.24	0.05	−0.18			0.41	
No.18	0.43	0.66	−0.05	−0.18	0.51	0.41	0.23	0.03	0.38	−0.15	−0.20	0.13	0.10	0.62			
No.14	0.29	0.66	−0.40	−0.21	0.41	0.11	0.25	0.01	0.45	−0.26	0.28	0.21	0.10			0.40	
No.12	0.45	0.64	−0.27	−0.21	0.43	0.05	0.02	0.29	0.41	−0.06	0.09	0.15	−0.04	0.48			
No.4	0.44	0.63	−0.31	−0.31	0.49	0.22	−0.10	0.34	0.29	−0.12	−0.21	0.23	−0.18		0.62		
No.7	0.24	0.58	−0.22	−0.09	0.18	0.17	−0.01	0.06	0.24	0.19	0.11	−0.01	0.18		0.46		
No.3	0.31	0.56	−0.38	−0.12	0.24	−0.07	0.08	−0.12	0.36	−0.06	0.25	−0.07	0.01	0.59			
No.23	0.16	0.43	−0.14	0.09	0.10	−0.11	0.09	−0.14	0.05	−0.12	−0.05	0.21	−0.35			0.70	
No.43	−0.07	−0.21	0.74	0.39	−0.25	−0.15	−0.09	0.00	−0.13	0.24	−0.14	−0.03	−0.07		0.71		
No.44	−0.06	−0.24	0.70	0.31	−0.22	−0.18	−0.01	−0.04	−0.06	0.08	0.00	0.14	−0.03			0.64	
No.46	−0.10	−0.14	0.68	0.38	−0.25	−0.26	0.01	0.04	−0.18	0.17	0.16	−0.11	−0.14	0.53			
No.42	−0.11	−0.16	0.66	0.31	−0.14	−0.19	−0.26	0.16	−0.21	0.57	−0.23	−0.26	0.02	0.42			
No.48	−0.17	−0.27	0.63	0.25	−0.17	−0.34	0.07	−0.09	−0.21	0.12	0.13	−0.08	−0.11			0.55	
No.34	−0.07	−0.20	0.63	0.23	−0.27	−0.19	0.12	0.02	−0.29	0.17	0.16	0.06	−0.02			0.43	
No.35	0.05	−0.17	0.25	0.72	−0.08	−0.16	0.20	−0.08	0.12	0.03	0.22	−0.09	0.21	0.44			
No.5	−0.19	−0.17	0.43	0.66	−0.15	−0.03	0.05	0.07	−0.16	0.00	−0.08	0.01	−0.11	0.67			
No.13	−0.05	−0.12	0.21	0.65	−0.03	−0.33	0.03	0.08	−0.13	0.11	−0.05	0.19	−0.19		0.66		
No.8	−0.14	−0.17	0.27	0.65	−0.20	−0.25	0.05	0.25	−0.25	0.07	0.21	−0.03	0.14				0.48
No.20	0.02	0.00	0.15	0.59	−0.10	−0.06	0.20	0.13	−0.21	0.27	−0.10	−0.10	−0.17	0.61			
No.15	−0.08	−0.12	0.33	0.49	−0.02	−0.26	0.10	−0.04	−0.31	0.38	0.18	−0.08	−0.23	0.59			
No.40	0.22	0.30	−0.21	−0.07	0.68	0.24	0.19	0.05	0.17	0.00	0.05	0.15	0.10	0.65			
No.41	0.23	0.31	−0.28	−0.14	0.65	0.06	0.08	−0.08	0.00	0.01	0.00	0.15	−0.03			0.64	
No.45	0.34	0.35	−0.15	−0.11	0.63	0.22	0.10	−0.13	0.36	−0.22	0.13	−0.02	−0.06	0.48			
No.32	0.26	0.40	−0.32	0.01	0.61	−0.10	0.13	0.06	0.07	−0.09	0.10	0.36	−0.02		0.70		
No.38	0.22	0.37	−0.04	−0.16	0.58	0.15	0.10	0.24	0.45	−0.14	−0.05	−0.21	−0.01				0.46
No.39	−0.11	−0.26	0.21	0.33	−0.22	−0.69	−0.01	−0.09	−0.22	0.26	0.06	−0.02	0.06			0.74	
No.47	0.05	−0.09	0.40	0.13	−0.07	−0.65	−0.06	0.26	−0.04	0.05	−0.02	−0.10	−0.01		0.52		
No.49	0.21	0.35	−0.24	−0.12	0.30	0.62	0.04	0.05	0.27	−0.25	0.19	0.16	0.18				0.70
No.50	0.22	0.16	0.00	0.09	0.12	0.03	0.68	0.14	0.07	0.07	−0.09	−0.04	0.02	0.40			
No.27	0.49	0.17	0.05	0.09	0.38	−0.09	0.55	−0.01	0.37	0.07	−0.02	−0.02	−0.07		0.44		
No.33	0.11	0.02	0.18	0.32	0.03	0.02	0.12	0.66	0.02	0.11	−0.06	−0.07	0.01	0.55			
No.29	0.25	0.44	0.03	0.11	0.33	0.25	0.09	−0.44	0.27	−0.16	−0.04	0.05	0.04		0.57		
No.10	0.18	0.15	−0.04	0.25	−0.10	0.01	0.38	0.43	0.22	−0.04	0.20	−0.18	0.34	0.52			
No.17	0.25	0.32	−0.11	−0.09	0.14	0.12	0.09	0.03	0.69	−0.07	0.00	0.07	0.00			0.56	
No.2	0.33	0.51	−0.11	−0.25	0.44	0.33	0.26	−0.29	0.53	−0.29	0.09	0.13	−0.11	0.68			
No.1	0.15	0.05	0.14	0.08	0.09	−0.10	0.14	0.07	0.02	0.69	0.03	0.06	−0.04			0.60	
No.16	−0.12	−0.23	0.55	0.12	−0.17	−0.24	−0.30	0.32	−0.21	0.57	−0.23	−0.26	0.03	0.61			
No.36	−0.03	−0.01	0.08	−0.02	0.04	0.10	−0.06	−0.05	0.02	0.00	0.60	0.03	0.02				0.82
No.51	0.15	0.21	−0.02	−0.06	0.19	0.15	0.02	−0.03	0.15	−0.03	0.01	0.68	−0.01	0.44			
No.37	0.32	−0.12	−0.20	−0.33	0.04	0.41	−0.18	0.10	0.37	0.13	−0.45	−0.49	0.08			0.48	
No.52	0.01	0.08	−0.18	−0.10	0.10	0.03	0.02	0.00	0.05	−0.05	0.01	0.03	0.73				0.65

Note: F1 = parents and the wider family; F2 = parents’ pessimistic attitude concerning their children; F3 = child’s characteristics; F4 = child’s physical incapacities.

**Table 4 ijerph-18-12174-t004:** Correlation between variables.

Category	QRS-F	CES-D	Problem Behavior Severity	Caregiving Time
QRS-F		0.281 **	0.300 **	0.212 **
CES-D			0.503 **	0.360 **
Problem behavior severity				0.737 **
Caregiving time				-

Note: QRS-F = Friedrich short form of the Questionnaire on Resources and Stress; CES-D = Center for Epidemiological Studies Depression; ** = *p* < 0.01.

## Data Availability

The data presented in this study are available on request from the corresponding author.

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
