# Peer review of "Validity of the Friedrich Short Form of the Questionnaire on Resources and Stress in Parents of Individuals with Autism Spectrum Disorder"

_ijerph, 2021, doi:10.3390/ijerph182212174_

Round 1
Reviewer 1 Report
Although the scientific soundness is good enough, there are some missing points than will clarify a better understanding of the article.
CFA and EFA analysis must be improved in order to clarify how the statistical analysis was developed. Questionnaire is not shown and should be better to have it to follow the explanation.
Finally, conclusions must be improved to reflect the true dimension of the analysis.
Author Response
Dear reviewer
Thank you very much for giving me the opportunity to resubmit my manuscript. I am grateful to you for insightful suggestions and comments. I have accepted or addressed all the suggested edits and comments. I hope that we have adequately strengthened out manuscript.
In the revised manuscript, I have highlighted in blue where we have made changes; please, note that I removed track changes that show all my edits.
The details of the changes are provided below this letter.
Thank you again for your time and efforts to review our manuscript and to provide insightful suggestions and edits.
Comment #1: Although the scientific soundness is good enough, there are some missing points than will clarify a better understanding of the article. CFA and EFA analysis must be improved in order to clarify how the statistical analysis was developed. Questionnaire is not shown and should be better to have it to follow the explanation.
Response #1: Statistical analysis was extended according to your comment (page 4, line 44 ~ page 5, line 5).
Comment #2: Finally, conclusions must be improved to reflect the true dimension of the analysis.
Response #2: Conclusions was revised to reflect the results of this study (page 8, line 44 ~line 47).
Reviewer 2 Report
|
Author Response
Dear reviewer
Thank you very much for giving me the opportunity to revise our manuscript. I am grateful to you for insightful suggestions and comments. I have carefully reviewed the comments and incorporated them to strengthen our manuscript.
In the revised manuscript, I have highlighted in blue where we have made changes; please, note that I removed track changes that show all my edits.
The details of the changes are provided below this letter.
Thank you again for your time and efforts to review my manuscript and to provide insightful suggestions and edits.
Comment #1: Table 2 header: it should be stated which instrument was subject of the factor analysis
Response #2: Title was revised (Table 2).
Comment #2: Results of rotated factor matrix are inevitable, and also graphical comparison with previous study (if I understood correctly by Holroyd [37]), this could be probably presented in a single table with 55 rows (header and 54 items) and approximately 18 columns (items description, 13 factors extracted in present study and 4 by Holroyd [37])
Response #3: Table 3 was inserted according to your comments (Table 3).
Comment #4: Also results should be described in a manner to understand what was analysed without the necessity for detailed search throughout the manuscript.
Response #4: Results was revised to show more detail description (page 5, line 36 ~ line 37; page 6, line 7 ~ line 12; page 7, line 17 ~line 20).
Reviewer 3 Report
There is insufficient knowledge about the psychometric properties of the Friedrich short form of the Questionnaire on Resources and Stress (QRS-F) used to measure the caregiving burden of caregivers of individuals with autism spectrum disorder (ASD). The present study, therefore, aimed to confirm the validity of the QRS-F. The existence of studies that deal with the development and improvement of assessment instruments aimed at caregivers is always positive. However, throughout the article there are certain shortcomings that need to be improved:
- Regarding the sample, to perform a factor analysis, the recommendations say that at least 10 people must be included per factor, so for a 54-item questionnaire, the sample should have been greater than 540.
- In the description of the participants, the types of ASD are not clarified, nor is it explained or justified, in epidemiological terms, the great gender difference between the participants.
- With what tool was the confirmatory factor analysis carried out? ? AMOS? Specify.
- Regarding the instruments used, the introduction describes the ZARIT, but then it is not applied as an instrument when calculating convergent validity and one that evaluates depression is used.
- The specific factors of the initial validation of the QRS-F are not described.
- In the introduction it is said that "is frequently used to study stress in families of children with disabilities [32] and also widely employed for measuring caregiving burden in families of children with ASD [33-36]. However, there is limited knowledge regarding its psychometric properties when measuring the burden 84 among caregivers of individuals with ASD ". Based on this, is it suggested that the referred studies are invalid?
- The exploratory factor analysis is not common, if the author has already done it, although the authors decide to carry out the confirmatory analysis since the exploratory does not have adjusted values. I think it would be pertinent to carry out a confirmatory analysis, but taking into account the sociodemographic characteristics. For example, considering the ASD subtypes separately (although they are not explained in the corresponding table). Or also taking into account the degrees of disability separately. The sample may be too heterogeneous to perform a single confirmatory analysis.
- The weights of the items must be considered since the inclusion or elimination of any of them can improve the results of the variance.
Author Response
Dear reviewer
Thank you very much for giving me the opportunity to revise our manuscript. I am grateful to you for insightful suggestions and comments. I have carefully reviewed the comments and incorporated them to strengthen our manuscript.
In the revised manuscript, I have highlighted in blue where we have made changes; please, note that I removed track changes that show all my edits.
The details of the changes are provided below this letter.
Thank you again for your time and efforts to review my manuscript and to provide insightful suggestions and edits.
Comment #1: There is insufficient knowledge about the psychometric properties of the Friedrich short form of the Questionnaire on Resources and Stress (QRS-F) used to measure the caregiving burden of caregivers of individuals with autism spectrum disorder (ASD). The present study, therefore, aimed to confirm the validity of the QRS-F. The existence of studies that deal with the development and improvement of assessment instruments aimed at caregivers is always positive. However, throughout the article there are certain shortcomings that need to be improved:
Response #1: Thank you for your review my manuscript. According to your helpful comment, manuscript was revised.
Comment #2: Regarding the sample, to perform a factor analysis, the recommendations say that at least 10 people must be included per factor, so for a 54-item questionnaire, the sample should have been greater than 540.
Response #3: Reference about the sample size was inserted and limitation related to sample size was stated (page 4, line 1 ~ line 6; page 8, line 32 ~ line 4).
Comment #3: In the description of the participants, the types of ASD are not clarified, nor is it explained or justified, in epidemiological terms, the great gender difference between the participants.
Response #3: The data was revised for sample method more detail page 3, line 10 ~ line 24).
Comment #4: With what tool was the confirmatory factor analysis carried out? ? AMOS? Specify.
Response #4: Yes. It was specified (page 5, line 23 ~ line 24).
Comment #5: Regarding the instruments used, the introduction describes the ZARIT, but then it is not applied as an instrument when calculating convergent validity and one that evaluates depression is used.
Response #5: The ZBS was one kind of scale such as Caregiver Strain Questionnaire and QRS-F which was described in introduction. However, the description about ZBS was too enough. The paragraph was revised (page 2, line 41 ~ line 47).
Comment #6: The specific factors of the initial validation of the QRS-F are not described.
Response #6: It was descried in measurement (page 4, line 10 ~line 14).
Comment #7: In the introduction it is said that "is frequently used to study stress in families of children with disabilities [32] and also widely employed for measuring caregiving burden in families of children with ASD [33-36]. However, there is limited knowledge regarding its psychometric properties when measuring the burden 84 among caregivers of individuals with ASD ". Based on this, is it suggested that the referred studies are invalid?
Response #7: This study verified the reliability and convergent validity of the QRS-F and the total score was recommended as the results of this study. So, it did not suggest that referred studies were invalid.
Comment #8: The exploratory factor analysis is not common, if the author has already done it, although the authors decide to carry out the confirmatory analysis since the exploratory does not have adjusted values. I think it would be pertinent to carry out a confirmatory analysis, but taking into account the sociodemographic characteristics. For example, considering the ASD subtypes separately (although they are not explained in the corresponding table). Or also taking into account the degrees of disability separately. The sample may be too heterogeneous to perform a single confirmatory analysis.
Comment #8: Confirmatory factor analysis was performed (See, page 4 ~ page 5).
Comment # 9: The weights of the items must be considered since the inclusion or elimination of any of them can improve the results of the variance.
Response #9: Rotated factor loading of each item was inserted (Table 3).
Round 2
Reviewer 1 Report
Most of the comments were attended (Grade of disability is not explained yet) and the paper seems to be right now to be understood even by a casual reader.
Author Response
Dear reviewer
Thank you very much for giving me the opportunity to resubmit my manuscript. I am grateful to you for insightful suggestions and comments. I have accepted or addressed all the suggested edits and comments. I hope that we have adequately strengthened out manuscript.
Thank you again for your time and efforts to review our manuscript and to provide insightful suggestions and edits.
Comment #1: Most of the comments were attended (Grade of disability is not explained yet) and the paper seems to be right now to be understood even by a casual reader.
Response #2: I described more detail regarding to grade of disability.
. In Korea, the diagnosis and grade of the disability by a doctor is needed for registration, Regarding to grade of disability, Grade 3 individuals with ASD those who were assessed to have a pervasive developmental disorder according to the diagnostic criteria of ICD-10 (International Statistical Classification of Diseases and Related Health Problems-10) that does not show the stage of normal development and has an IQ of 70 or less, and a Global Assessment Scale for Developmentally Disables (GAS) score of 20 or less due to impairment of function and ability were most common (39.6%). Grade 2 individuals with ASD those who were assessed to have a pervasive developmental disorder according to the diagnostic criteria of ICD-10 that does not show the stage of normal development and has an IQ of 70 or less, and GAS score of 21-40 due to impairment of function and ability were 33.1%. Grade 3 individuals with ASD those who were assessed to have a pervasive developmental disorder according to the diagnostic criteria of ICD-10 that does not show the stage of normal development and has an IQ 71 or above, and GAS score of 41-50 due to impairment of function and ability were 27.3%.
Reviewer 3 Report
The authors say: (p.5) "3.1. Confirmatory Factor Analysis
The four-factor model proposed in a previous study was verified for parents of individuals with ASD"
Which study are you referring to? I can't find it in the text.
Author Response
Dear reviewer
Thank you very much for giving me the opportunity to resubmit my manuscript. I am grateful to you for insightful suggestions and comments. I have accepted or addressed all the suggested edits and comments. I hope that we have adequately strengthened out manuscript.
Thank you again for your time and efforts to review our manuscript and to provide insightful suggestions and edits.
Comment #1: The authors say: (p.5) "3.1. Confirmatory Factor Analysis
The four-factor model proposed in a previous study was verified for parents of individuals with ASD"
Which study are you referring to? I can't find it in the text.
Response #1: I inserted specific reference and described more detail.
The QRS-F was developed by Holroyd [37] to measure the burden of caring for children with chronic diseases [31]. This 52-item scale measures the caregiving burden resulting from problems faced by parents. This scale consisted of four factor such as the wider family, parents’ pessimistic attitude concerning their children, child’s characteristics, and child’s physical incapacities, with item-total correlations ranging from 0.15 to 0.63 and a mean inter-item correlation of 0.26.
The four-factor model proposed in a previous study [37] was verified for parents of individuals with ASD.